# Correlation between Overconfidence and Learning Motivation in Postgraduate Infection Prevention and Control Training

**DOI:** 10.3390/ijerph19095763

**Published:** 2022-05-09

**Authors:** Milena Trifunovic-Koenig, Stefan Bushuven, Bianka Gerber, Baerbel Otto, Markus Dettenkofer, Florian Salm, Martin R. Fischer

**Affiliations:** 1Institute for Infection Control and Infection Prevention, Health Care Association District of Constance (GLKN), 78315 Konstanz, Germany; milenatriko@gmx.de (M.T.-K.); markus.dettenkofer@sana.de (M.D.); florian.salm@glkn.de (F.S.); 2Hegau-Jugendwerk Hospital Gailingen, Health Care Association District of Constance (GLKN), 78262 Konstanz, Germany; 3Institute of Medical Education, University Hospital, Ludwig Maximilian University of Munich, 80336 Munich, Germany; martin.fischer@med.uni-muenchen.de; 4Institute of Anesthesiology, Intensive Care, Emergency Medicine and Pain Therapy, Hegau-Bodensee-Hospital Singen, Health Care Association District of Constance (GLKN), 78224 Singen, Germany; bianka.siebold@gmail.com; 5Institute of Laboratory Medicine, University Hospital, Ludwig Maximilian University of Munich, 80336 Munich, Germany; baerbel.otto@med.uni-muenchen.de

**Keywords:** hand disinfection, infection control training, motivation for learning, overconfidence, nosocomial infection prevention

## Abstract

**Introduction:** Training in hand hygiene for health care workers is essential to reduce hospital-acquired infections. Unfortunately, training in this competency may be perceived as tedious, time-consuming, and expendable. In preceding studies, our working group detected overconfidence effects in the self-assessment of hand hygiene competencies. Overconfidence is the belief of being better than others (overplacement) or being better than tests reveal (overestimation). The belief that members of their profession are better than other professionals is attributable to the clinical tribalism phenomenon. The study aimed to assess the correlation of overconfidence effects on hand hygiene and their association with four motivational dimensions (intrinsic, identified, external, and amotivation) to attend hand hygiene training. **Methods:** We conducted an open online convenience sampling survey with 103 health care professionals (physicians, nurses, and paramedics) in German, combining previously validated questionnaires for (a) overconfidence in hand hygiene and (b) learning motivation assessments. Statistics included parametric, nonparametric, and cluster analyses. **Results:** We detected a quadratic, u-shaped correlation between learning motivation and the assessments of one’s own and others’ competencies. The results of the quadratic regressions with overplacement and its quadratic term as predictors indicated that the model explained 7% of the variance of amotivation (*R*^2^ = 0.07; *F*(2, 100) = 3.94; *p* = 0.02). Similarly, the quadratic model of clinical tribalism for nurses in comparison to physicians and its quadratic term explained 18% of the variance of amotivation (*R*^2^ = 0.18; *F*(2, 48) = 5.30; *p* = 0.01). Cluster analysis revealed three distinct groups of participants: (1) “experts” (*n*_1_ = 43) with excellent knowledge and justifiable confidence in their proficiencies but still motivated for ongoing training, and (2) “recruitables” (*n*_2_ = 43) who are less competent with mild overconfidence and higher motivation to attend training, and (3) “unawares” (*n*_3_ = 17) being highly overconfident, incompetent (especially in assessing risks for incorrect and omitted hand hygiene), and lacking motivation for training. **Discussion:** We were able to show that a highly rated self-assessment, which was justified (confident) or unjustified (overconfident), does not necessarily correlate with a low motivation to learn. However, the expert’s learning motivation stayed high. Overconfident persons could be divided into two groups: motivated for training (recruitable) or not (unaware). These findings are consistent with prior studies on overconfidence in medical and non-medical contexts. Regarding the study’s limitations (sample size and convenience sampling), our findings indicate a need for further research in the closed populations of health care providers on training motivation in hand hygiene.

## 1. Introduction

The prevention of hospital acquired infections (HAI) is a key component of patient safety [1,2,3]. Hand hygiene training is essential to establish and maintain infection prevention and control (IPC) competencies of health care workers (HCWs) [4,5]. Although programs that aim to increase IPC competencies among HCWs have been broadly promoted and supported, training in hand hygiene is mostly perceived to be tedious, unstructured, or expendable [6,7] and may fail to substantially reduce nosocomial infection rates [8]. However, there is no clear concept of specific factors contributing to learning motivation in this context. The individually perceived necessity for training attendance and self-assessment of the hand hygiene competencies might play a critical role in the motivational process. 

However, self-assessment is known to be a poor predictor of measurable knowledge and observable skills [9]. Biased by flawed self-assessment, HCWs may overestimate their skills of easy tasks or tasks perceived to be easy. These effects relate to the phenomenon of the *overconfidence effects* [9], documented in almost every area of life [10,11,12,13]. They can be subdivided into: overestimation, the belief of being better than objectively measured; overplacement, the belief of being better than others; and overprecision, the belief of being very accurate in their self-assessment [13].

In hand hygiene, recent findings of our research group suggest that medical under- and postgraduates and local and national HCWs rate themselves superior compared to colleagues and supervisors [7]. The additional effect, overrating their own profession above others, is attributable to the *clinical tribalism phenomenon* [14]. 

In this study, we postulated that overconfidence might impact learning motivation. Based on previous findings of educational studies [15], we assumed that the association between overconfidence and motivation is negatively linear, i.e., high levels of overconfidence are related to reduced motivation: “*If I am the best-trained person, why should I attend a time-consuming and maybe boring training on hand hygiene?*” In addition, hand hygiene compliance is connected to patients’ safety factors, such as the safety culture, and the supervisors’ perceptions about the relevance of hand hygiene in their departments and units [16,17]. Thus, we explored the role of the perceived risk to the patient after not performing the indicated hand hygiene procedure in the relationship between overconfidence and motivation.

This study was based on the self-determination motivational theory [18,19,20]. This theory postulates that motivation can be divided into two distinct groups: intrinsic (“*I want to learn*”—the stimulus comes from the individual), and extrinsic motivation (“*I have to learn*”—the stimulus comes from the environment) [18,19,20]. Extrinsic motivation can further be divided into subgroups (e.g., external regulation, identified regulation, etc.). Intrinsic motivation drives enthusiasm and performance [21]. External regulation, in contrast, usually leads to poor performance and decreased well-being in the long term [22].

This study aimed to investigate the effects of overconfidence on HCWs’ motivation to participate in IPC training. To examine this impact, we analyzed the correlations between overplacement, clinical tribalism, and motivation. In addition, we examined the impact of overestimation on motivation and their relative dependence on the subjectively perceived risk assessment in distinct groups of HCWs. 

In this work, we hypothesized that:

Motivation to attend hygiene training (measured by the Situational Motivational Scale (SIMS)) depends on self-assessment and different overconfidence effects (overestimation, overplacement, and clinical tribalism) as measured by the SATIS-4D questionnaire [7]:

SIMS-dimensions are correlated in a linear (H1a) or quadratic (H1b) manner with:

**Hypothesis** **1a** **(H1a):**
*The absolute value of self-assessment in different infection control proficiencies.*


**Hypothesis** **1b** **(H1b):***Overplacement in infection control as a difference of assessing one-self and others in different infection control proficiencies*.

**Hypothesis** **2a** **(H2a):**
*The diverse population of HCWs can be grouped based on their overestimation and the learning motivation, i.e., on the following indicators:*
-
*HCW’s self-assessment, self-reported competencies, and knowledge of infection control methods;*
-
*Levels of the four SIMS-dimensions; and*
-
*Self-evaluation of adherence to the 5 WHO Hand Hygiene indications in terms of percentage.*



**Hypothesis** **2b** **(H2b):***Identified subgroups of HCWs differ in their view of patient safety*.

We realize that all parameters are based on self-assessment rather than actual measured data. We decided to do so because the motivation to attend a voluntary training is based on this subjective self-assessment and not on objective pretesting. It is not our intention to test whether the incentive based on self-assessment is ethically justifiable or not. Moreover, it is part of our project to add data to this topic to raise further qualitative and ethical questions. 

## 2. Methods

### 2.1. Study Design and Variables

We combined the German version of the SIMS [23] and parts of the originally validated German SATIS-4D questionnaires (subjective assessment of training in infection prevention skills) [7] to form the questionnaire available with the relevant items for the present study in the German version with English translation in the Appendix A. The questionnaire provides a wide range of items, including demographic data (gender, calendar age, subjective age, profession, educational course, and workplace), motivation, self-assessment of own proficiencies, assessments of colleagues, and competencies of physicians, nurses, and paramedics. Furthermore, the estimation of potential harm to patients and occurrence of these events, self-assessments of their own behavior in different situations concerning hand hygiene per WHO standards, the self-assessment of speaking-up behavior, and observation of colleagues and other HCWs on speaking-up and constructive intervention in case of an error in hand hygiene were reported. Finally, a free text entry was included in the questionnaire for further comments. 

For this first approach, we aimed for a minimum of 100 participants to complete the survey. 

### 2.2. Measurements 

*Motivation* was measured with the pretested German translation of the SIMS instrument and adjusted for the setting of infection control training [23]. The participants were asked to reflect on the reasons for attending a typical infection control training. The SIMS instrument is divided into four subscales. Each of the subscales consists of four items: intrinsic motivation (*Cronbach’s alpha* in this study α = 0.78, e.g., “*Because I feel good when doing this activity*”), identified regulation (α = 0.80, e.g., “*Because I am doing it for my own good*”), extrinsic regulation (α = 0.73, e.g., “*Because I am supposed to do it*”), and amotivation (α = 0.86, e.g., “*I don’t know; I don’t see what this activity brings me*”). All items are scored on an ascending 5-point Likert Scale ranging from 1 “*completely disagree*” to 5 “*completely agree*”.

*Self-Assessment*. We used the six-items-instrument (α = 0.80) of the SATIS Questionnaire to measure the self-assessment of proficiencies in IPC. The scale measures the following aspects of the proficiencies: three items address the situational application of the knowledge in hygienic hand disinfection (e.g., “*I*
*conduct hand hygiene if indicated in a situation*”), one item addresses the provision of feedback (“*I identify mistakes in the conduction of hand hygiene in other persons*”), one item addresses feedback reception (“*Depending on the situation I accept feedback if others correct me for an error in hand hygiene*”), and one item addresses the speaking up behavior after incorrect hand disinfection (“*I correct others readily if I perceive a mistake in hand hygiene*”). 

*Assessment of other professionals.* We used the 5-item-instrument for the assessments of colleagues of the same profession (α = 0.78) and the three-item-instrument of the SATIS Questionnaire for each of the three professional groups (physicians α = 0.78, nurses: α = 0.72, and paramedics α = 0.78). The scales included the same aspects as the self-assessments: the situational application of knowledge in hygienic hand disinfection (e.g., “*My colleagues conducts hand hygiene if indicated in a situation*”), feedback reception (“*Depending on the situation, nurses accept feedback if others correct them for an error in hand hygiene*”), and speaking up after incorrect hand disinfection (“*Physicians correct others readily if they perceive a mistake in hand hygiene*“). However, the provision of feedback was not included. 

These assessments were similarly scored as the items of SIMS on a five-point rating scale, ranging from 1 “*completely disagree*” to 5 “*completely agree*”.

*Overplacement Effect.* The overplacement-score was computed as an index of the differences between the corresponding items of assessing oneself and others (e.g., the values of the item “*Depending on the situation, my colleagues accept feedback if others correct them for an error in hand hygiene*” were subtracted from the values of the item “*Depending on the situation, I accept feedback if others correct me for an error in hand hygiene if indicated in a situation*”).

*Clinical Tribalism Effect.* Analog to overplacement, the clinical tribalism score was computed as an index of the differences between the corresponding items of assessing oneself and the competencies of other HCW profession groups. 

*Self-reported hand hygiene compliance* was assessed with an instrument consisting of 12 different items. The participants were asked to specify how often they perform the hand disinfection in nine indicated situations typically encountered in daily routine (one-time WHO 3, two-times WHO 1, 2, 4, and 5) in accordance with the WHO Hand Hygiene guidelines: “In the following situations I conduct hygienic hand disinfection”: After moving a used patient bed (WHO 5)After contaminating your hand with urine (WHO 3)Before connecting an infusion to an IV line (WHO 2)After shaking hands with a patient (WHO 4)Before connecting a urinary catheter to a collection bag (WHO 2)After helping a patient up after a fall (WHO 4)Before shaking hands with a patient (WHO 1)Before positioning a patient on an operation table (WHO 1)After picking up a patient’s towel that has fallen to the ground in the lavatory (WHO 5)

Additionally, we added three distractor situations not explicitly indicated by the 5 WHO Hand Hygiene guidelines (e.g., “*after preparing sterile IV medication*”). Hence, these items tested factual and situational knowledge and self-assessment. The rating procedure is defined as follows, with higher ratings pointing to higher occurrence: (1) never (<1%), (2) seldom (1–25%), (3) sometimes (25–75%), (4) commonly (75–99%), and (5) always (>99%). In addition, we computed separate index-scores for indicated situations (α = 0.83) and non-indicated situations (α = 0.71).

*Overestimation Effect.* We descriptively compared the self-assessment score of one’s own proficiencies in infection control and the two scores of indicated and non-indicated scenarios for hand disinfection by the WHO protocols of self-reported hand hygiene compliance to estimate the level of overestimation across the identified clusters.

*Patients’ safety*. We used two separate items to measure the self-estimated degree of patients’ safety according to international standards of ISO 31000 of medical failure mode effect analysis (FMEA). First, the potential harm to patients was estimated: “The maximum credible effect of omitted hand hygiene is without consequence (1), minor—without any long-lasting effect (2), severe—with a prolonged hospital stay (3), critical—with long-lasting effects (4), and lethal (5)”. The second item addressed the observed occurrence of these events in one’s own clinical experience: “How often is a patient harmed in your working environment?” Uncommon (1x > 3 years) (1), seldom (once every 3 years) (2), moderate (once a year) (3), often (once every 3 months) (4), and very often (once a month) (5). The items are based on the ISO 31000 scales in risk management [24]. 

### 2.3. Setting and Participants

After completing the pretesting and reevaluation, and after receiving approval from the ethical committee of the physicians’ association of Stuttgart, we distributed the questionnaire in the German language online to more than 5000 HCWs via social media platforms (e.g., professional groups of nurses, paramedics, and physicians with defined numbers of memberships), their own networks, and the associated institutions from July to October 2019. Laypersons were not addressed. We decided to address the whole spectrum of HCWs with different professions to achieve a deeper insight into subgroups and differences between professions. This broad dissemination was conducted to limit selection bias and increase response rates. However, this also created other biases, as discussed later in this manuscript [25]. Of more than 5000 persons addressed in these social media groups, 135 persons responded to the questionnaire, of whom 93 completed the survey entirely. An additional 10 participants only partially completed the survey, but reported all the relevant items to be included in the study. In total, the sample consisted of 103 HCWs. Participants were repeatedly informed about the study on social media boards (professional and thematic boards) or by e-mail (newsgroups). The relevant items for the present study are shown in Appendix A altogether with an English translation, Appendix A) pointing out the study objectives, participation guidelines, as well as a statement on publication, ethics, and anonymity. 

We recorded the values of SIMS, self-assessment of own proficiencies, self-assessment of the same and other professional groups_,_ self-reported hand hygiene compliance, and patients’ safety on an ascending 5-point Likert Scale. The higher values were assigned to the higher expressions or occurrence of the construct/phenomena for easier interpretation and understanding of the intercorrelations. Hence, we deviated from the original range applied in the validation studies.

### 2.4. Statistical Methods

We tested the first hypothesis (H1a) by inspecting Pearson’s product-moment correlation matrix with an additional bootstrapped estimation of 95% confidence intervals for correlation coefficients (bias corrected and accelerated method (*BCa*) based on 1000 samples). We tested hypothesis H1b by performing quadratic regressions where each of the SIMS-dimensions served as a criterion variable in combination with the following variables: Overplacement, Clinical tribalism, and their corresponding quadratic terms which were used as predictors in corresponding regression equations. We centered (weighted by the mean values) the predictor variables prior to conducting regression analyses in order to avoid multicollinearity [26], which is a common problem in quadratic regressions due to strong correlations between the variable and its quadratic term. To ensure robust estimation of the regression coefficient, we applied bootstrapping (bias-corrected and accelerated method, based on 1000 samples) when homoscedasticity and normal distribution of residuals were not present [27,28]. 

We tested the second hypothesis H2a by performing a K-means cluster analysis to establish appropriate homogenous subgroups within the more complex heterogeneous population [29]. All variables included in the cluster analysis were continuous, measured with the analog ascending 5-point Likert Scale. In addition, we inspected the intercorrelation matrix of the variables we wanted to include in the cluster analysis and consequentially excluded the variables with a Pearson correlation coefficient higher than 0.5 [30]. Considering that the statistical power of the cluster analysis depends on the defined number of clusters and the number of the variables which are included in the cluster analysis [31], we ran the cluster analysis for the *k* = 2 to *k* = 10 cluster numbers. The number of groups was determined by reviewing the sum of squared errors (SSE) and the calculated Silhouette value compared to the cluster numbers. A lower SSE and higher Silhouette value indicated a sharper separation between the groups [32]. 

We tested the hypothesis H2b using two independent one-way Analyses of Variance (ANOVA). The group variable was taken from the cluster value the group belonged to and the dependent variables were the two items measuring patients’ safety. To obtain robust confidence intervals of the differences of means between the clusters in the post-hoc contrast analyses, we performed bootstrapping (bias corrected and accelerated *BCa*, based on 1000 samples) when we were not satisfied with the assumptions required to perform ANOVA; for example, in the case of heterogeneity of variances [33]. Before conducting one-way ANOVAs, we conducted post-hoc power analyses to ensure we had a sufficient sample to detect large effects. The reasoning behind conducting the power analyses at this point was that we were not familiar with the number of clusters we should expect after performing the K-means cluster analysis due to the explorative character of the present study. 

We tested possible differences in age distributions between identified clusters using the Independent-Samples Kruskal-Wallis Test with Bonferroni correction. Differences in gender proportions, as well as proportions of participants working on different health care levels between the detected clusters, were analyzed using the Chi^2^ test. 

Statistical analyses were performed using SPSS Software Version 27 (IBM SPSS Statistics for Windows, Version 27.0. Armonk, New York, NY, USA) and G* Power Version 3.1 (G*Power, Version 3.1, Faul et al., Heinrich Heine University of Düsseldorf, Düsseldorf, Germany) [34,35].

## 3. Results

### 3.1. Description of the Sample

A total of 103 participants were included in the study. The sample comprised 21 (20.4%) male, 79 (76.7%) female, and one (1%) divers/transgender/trans identical participants. The sex ratio (male to female) of the sample was very close to that of the population of HCWs in Germany (75.6% female) [36]. The mean age was 36.43 years with a standard deviation of 17.31 years. Nearly half of the participants were nurses (49.5%), followed by physicians (11.7%) and paramedics (3.2%). More than one-third of all participants (35%) reported being part of another professional group not listed in the questionnaire. According to the social media boards addressed, these HCWs were most likely speech and language therapists, physiotherapists, and psychologists. Considering that the percentage of physicians compared to nursing staff (including paramedics and midwives) in Germany is approximately 35%, the physicians were clearly underrepresented in the sample [37]. A total of 93.2% of the participants had finished their professional training (post-graduates). Among them, 15 participants worked as educators. Only four participants were still undergoing training at the time (e.g., nursing or medical students), and further analysis for this very small group was not conducted. Almost 70% of the participants worked in hospitals (16.5% in primary, 35% in secondary, and 18.4% in tertiary health care). Table 1 describes the sample characteristics.

### 3.2. Hypotheses Testing 

#### 3.2.1. H1a and H1b: Correlations between Overplacement, Clinical Tribalism, and Amotivation

Table 2 shows the Pearson’s product-moment correlation matrix with 95% *BCa* confidence intervals for correlation coefficients between the four SIMS Subscales: self-assessment of own proficiencies, overplacement, clinical tribalism, and self-reported hand disinfection compliance for indicated as well as not indicated situations for hand disinfection. 

According to the matrix, only four SIMS subscales were correlated to one another. In contrast, the nurses’ self-assessment of their proficiencies, overplacement, and clinical tribalism compared to the nurses’ assessed proficiencies of physicians and paramedics were not correlated to any of the SIMS subscales. Please note that we could only compute the values of clinical tribalism for nurses due to the small number of physicians and paramedics in the sample.

In sum, our hypothesis H1a, which postulated that overconfidence linearly correlates with learning motivation, should be rejected.

Two aspects of overconfidence—clinical tribalism and overplacement—were considered in relation to four forms of motivation: intrinsic, external, identified regulation, and amotivation. Both overplacement and clinical tribalism showed no significant quadratic associations with intrinsic motivation, identified, and external regulation (see Appendix A). The only significant quadratic relationships were overplacement and amotivation (*R*^2^ = 0.07; *F*(2, 100) = 3.94; *p* = 0.02), whereas the quadratic term of overplacement significantly predicted amotivation (*B* = 0.16; 95% *BCa* [0.01–0.26]; *p* = 0.03). The correlation had a U-form (see Figure 1). 

Correspondingly, an equivalent significant quadratic correlation was found between clinical tribalism for nurses in comparison with physicians and amotivation (*R*^2^ = 0.18; *F*(2, 48) = 5.30; *p* = 0.01). The quadratic term of clinical tribalism significantly predicted amotivation (*B* = 0.52; 95% *BCa* [0.02–0.83]; *p* = 0.01). This correlation also had the U-form (see Figure 2). In contrast, there was no significant quadratic regression between clinical tribalism (a difference between nurses’ self-assessment and nurses’ perception of paramedics’ competencies) and amotivation. 

Thus, hypothesis H1b was partially confirmed.

In sum, the findings suggest that overconfidence does not linearly correlate with motivation. Furthermore, the graphical presentation of the relationship (U-form) between overplacement with amotivation shows three different ranges of overplacement with different associative patterns to motivation: (1) extremely underplaced participants with a high level of amotivation, (2) extremely overplaced participants with a high level of amotivation, and (3) participants between these extremes with a low level of amotivation. 

#### 3.2.2. H2a: K-Means Cluster Analysis

Prior to conducting the K-means cluster analysis, we examined the Pearson’s correlation matrix, including four SIMS dimensions, self-assessment, indicated, and not indicated situations for hand hygiene (Table 2). Of the four SIMS dimensions, we decided to exclude identified regulation due to its strong intercorrelations with amotivation and intrinsic motivation. Therefore, we proceeded with self-assessment, intrinsic motivation, external regulation, and amotivation. Additionally, we excluded the non-indicated situations for hand disinfection due to its strong intercorrelation with the indicated situations. Altogether, five variables were included in the K-means algorithm: (1) self-assessment, (2) intrinsic regulation, (3) extrinsic regulation, (4) amotivation, and (5) indicated situations for hand hygiene.

In the next step, we performed the K-means cluster analysis for the *k* = 2 to *k* = 10 numbers of clusters. In addition, we compared the SSE and the average Silhouette coefficients with the number of clusters. The three-cluster solution showed the highest mean Silhouette value of 0.303 (see Appendix A). Furthermore, the plot of the SSE showed an elbow at *k* = 3, indicating that three is the optimal number of clusters (elbow method, see Appendix A). Hence, we concluded that the three-cluster solution was the most appropriate. Therefore, hypothesis H2 was confirmed.

#### 3.2.3. H2b: External Validation of the Three-Cluster Solution: One-Way ANOVAs

Prior to conducting the one-way ANOVA, we performed post-hoc power analyses to ensure that the small sample size does not provoke a type 2 error hindering the significance of the results. ANOVA (fixed effects, omnibus, one-way) with the large effect size of *f* = 0.40, α *err prob* = 0.05, power (1 − β *err prob*) = 0.80, shows three independent groups, numerator *df* = 2; denominator *df* = 63 is the actual statistical power 0.82, and the optimal sample is size *n* ≥ 66 (see Appendix A). Therefore, our sample size was able to detect large possible effects.

The first ANOVA was conducted with the item: “The maximum credible effect of omitted hand hygiene is: without consequence (1), minor—without any long-lasting effect (2), severe—with a prolonged hospital stay (3), critical—with long-lasting effects (4), and lethal (5)”, as the dependent variable, and the number of the clusters’ membership as the factor variable. The one-way ANOVA determined a statistically significant difference between groups (*F*(2, 90) = 5.73, *p* = 0.004). The Bonferroni post-hoc test revealed that the maximum credible effect of omitted hand hygiene was estimated to be statistically significantly higher by the members of the first cluster member (*M*1 = 4.16; *SD* = 0.90) as well as by the members of the third cluster (*M*3 = 4.20; *SD* = 0.85), in comparison with the second cluster (*M*2 = 3.38; *SD* = 0.81). Table 3 shows the results of the Bonferroni post-hoc comparisons. Eta-squared indicated the medium effect size as 0.11. Figure 3 shows a boxplot of the dependent variable in the first ANOVA for the three clusters separately.

The second ANOVA was conducted with the item: “How often is a patient harmed in your working environment? Uncommon (once in more than 3 years) (1), seldom (once every 3 years) (2), moderate (once a year) (3), often (once every 3 months) (4), and very often (once a month) (5)”. In contrast to the first analysis of variance, the second one-way ANOVA revealed no statistically significant difference between the groups (*F*(2, 73) = 0.17; *p* = 0.84). Appendix A shows a boxplot of the dependent variable in the second ANOVA for the three clusters separately.

### 3.3. Description of the Clusters

We used the mean cluster values of the self-assessment scale, four SIMS dimensions, and the item measuring the aspect of the patients’ safety as the maximal risk of harm to the patient after incorrect hand disinfection for the description of the clusters. According to the WHO recommendations, performing hand hygiene procedures in each indicated situation is unrealistic and, therefore, is not explicitly requested [8]. However, there is no global established maximum tolerance to hand hygiene non-compliance. Most scholars argue that a hand hygiene compliance of 75–80% of all requested hand hygiene moments should be considered satisfactory in clinical practice [38,39]. Thus, we considered a minimum hand hygiene compliance of 75% as acceptable, i.e., we postulated that hand hygiene should be carried out in at least 75% of cases when the guidelines require it. With this in mind, we compared the mean values of the self-reported frequencies for performing hand disinfection in indicated and not-indicated situations across the clusters with the value of four (75%) to describe the self-reported hand hygiene compliance. Thus, participants had to identify the indication (factual and situational knowledge) for each given hand hygiene scenario and assess their behavior. Appendix A shows the characteristics of the clusters. 

#### 3.3.1. Cluster 1—“Experts”

A total of 43 HCWs were grouped in the first cluster. Participants in this cluster reported high confidence levels in their proficiencies (*M* = 4.18; *SD* = 0.45). When given practical scenarios of when a hand hygiene procedure was required or not, experts exceeded the 75% compliance recommendation. However, in scenarios when it was not required, this cluster carried out hand hygiene less than 75% of the time. The connection between participants’ factual and situational knowledge and their high levels of adherence to WHO hand hygiene protocols explain the low levels of overestimation in this cluster. Amotivation was the lowest in this cluster (*M* = 1.30; *SD* = 0.83), and their recognition of the high risk to patients when the protocol is not followed (*M1* = 4.16; *SD* = 0.89) corresponds with the high levels of intrinsic motivation (*M* = 3.74; *SD* = 0.64) and identified regulation (*M* = 4.41; *SD* = 0.45). Although competence was the highest in this cluster, attending training remains an essential aspect of the participants’ perceptions of professional duty to maintain high competence and expand knowledge when new policies are introduced.

#### 3.3.2. Cluster 2—“Unawares”

A total of 17 HCWs were grouped in the second cluster. Participants in this cluster highly ranked their confidence in their competencies (*M* = 4.19; *SD* = 0.72). However, their knowledge of the hand hygiene protocol and the risk that non-compliance poses to patients were significantly lower than in the first cluster (*M*2 = 3.35; *SD*2 = 1.11). Unawares reported that hand hygiene was required in all scenarios presented to them, although this was only required in some. In contrast to the other two clusters, these participants perceived the risk to patients as relatively low. On average, option 3 (a prolonged hospital stay without causing permanent damage) was chosen as the item addressing potential risk for patients after non-compliance with the protocol. Previous research confirms that awareness of patient safety motivates learning [16]. Thus, the lack of knowledge and understanding of this risk may explain this clusters’ relatively high amotivation levels (*M* = 3.10; *SD* = 0.71) and especially low intrinsic motivation (*M* = 2.04; *SD* = 0.59). Although participants in this cluster would benefit the most from further training, their amotivation to attend training presents a significant obstacle to learning. This places patients at risk for nosocomial infection. 

#### 3.3.3. Cluster 3—“Recruitables”

A total of 43 HCWs were grouped in the third cluster. Overestimation in the third cluster was due to discrepancies between participants’ reported confidence levels and their failure to comply with the hand hygiene protocol. These participants reported lower confidence levels in their knowledge and practical compliance than those in the other two clusters (*M* = 3.78; *SD* = 0.56). However, when presented with various scenarios, their reported theoretical and practical compliance with WHO guidelines was below 75%. In addition, the average values of both intrinsic (*M* = 3.08; *SD* = 0.63) and external motivation (*M* = 3.42; *SD* = 0.57) to learn were moderate in this cluster; thus, amotivation was low (*M* = 1.48; *SD* = 0.49). Low levels of amotivation could be explained by the participants’ responsibility to reduce the risk to patients, as recruitables perceived a high risk of harm when the hand hygiene protocol was not followed (*M3* = 4.20; *SD* = 0.85). Furthermore, these participants were likely to make a cognitive connection between training attendance and improvement in IPC.

There was no difference in the age distribution between the clusters (Independent-Samples Kruskal-Wallis Test, Kruskal–Wallis test with Bonferroni correction H(2) = 1.16; *p* = 0.56). A chi-square test of independence showed that there was no significant association between neither gender (X^2^ (2, 101) = 2.51, *p* = 0.64) nor workplace (ambulant, primary, secondary, or tertiary hospital level) (X^2^ (4, 79) = 9.15, *p* = 0.17) with the cluster membership. 

## 4. Discussion

This study demonstrated that overplacement and clinical tribalism as overconfidence effects were present and related to motivation to attend IPC training in this interprofessional sample of HCWs. Nevertheless, both quadratic regressions and K-means cluster analysis suggest that the relationship between these variables is not linear, i.e., a higher self-assessment than others and self-assessed hand hygiene compliance are not always related to poorer motivation levels to attend IPC training. In addition, in line with previous findings, our study demonstrates that their relationship is much more complex and influenced by other variables, such as perceived patients’ safety aspects [16]. Moreover, our findings suggest that it seems preferable to examine these relationships within distinct subgroups of HCWs instead of the whole population.

### 4.1. Hypothesis H1a and H1b

The hypothesis H1a should be rejected since we obtained no linear correlation between overconfidence dimensions (overplacement and clinical tribalism) and motivation. In contrast, hypothesis H1b was partially confirmed since we could identify statistically significant associations between overplacement and clinical tribalism and amotivation. Specifically, the relationship was quadratic (u-shaped): only extremely over- or underconfident participants were amotivated, whilst intermediately overconfident persons showed low levels of amotivation. The highlighted relationships between overplacement and clinical tribalism and amotivation may be explained by the unique definition of amotivation as an absence of motivation (no regulation), which is in contrast to all other motivation forms where motivation is present and defined by internal or external factors [18,19]. Amotivation is described in the literature as an excessive motivational dimension, leading to resignation and poor performance [40,41]. In our study, amotivation appears to stem from participants not cognitively associating the action (attending infection control training) and the consequence of inaction (spread of diseases), which significantly impacts patient safety in terms of IPC. However, unlike other motivation forms, amotivation as a variable poses increased challenges for the linear statistical analyses. Amotivation is usually empirically characterized by a skewed frequency distribution for the self-reported questionnaire data leading to difficulties obtaining linear correlations with other variables [42]. In this study, amotivation, measured by the SIMS instrument, also demonstrated a skewed distribution which could be the obstacle to detecting significant linear correlations between overplacement and amotivation.

Hence, the small percentage of variance explained by quadratic terms of overplacement and clinical tribalism indicates that the variables included in the quadratic regression model are insufficient to explain learning motivation in this context. Subjective risk assessment for the patients, as a consequence of non-compliance with the guidelines, appears to be a relevant variable connected to overconfidence and motivation and therefore should also be included in the model as indicated by the result of the cluster analysis and one-way ANOVA (H2b).

Amotivation, as the only motivational facet, appears to be related to overplacement. Furthermore, the efforts to reduce amotivation seem to be essential, considering the consequences that amotivation can provoke [22]. Therefore, all causes of amotivation in this context should be thoroughly examined, e.g., the correlation of amotivation with workload [43], safety culture [9], or even working hours [44]. In addition, overplacement appears to be a contributing factor. Hence, the analysis of the association between these two constructs and further factors deserves special attention in future research.

### 4.2. Hypothesis H2a and H2b

The hypothesis H2a was confirmed. The K-means cluster analysis yielded three different clusters based on overestimation and motivation. Therefore, the sample of HCWs could be grouped into distinct clusters. In contrast, the participants within the clusters showed similar correlation patterns, and the participants between different clusters showed differing correlations between overestimation and learning motivation. The hypothesis H2b was also confirmed. Participants in the second cluster (Unawares) perceived the aspects of patients’ safety as significantly lower than the participants in the other two clusters 

Hence, the population of HCWs is relatively heterogeneous as there is no unique way to describe correlations between overconfidence and motivation. Nonetheless, we believe we found a method to deal with the heterogeneity by classifying the health care workers into distinct homogeneous groups and by studying these correlations in each of the groups separately. In this way, we obtained different profiles of the participants regarding overconfidence (overplacement, clinical tribalism, overestimation), self-assessed risk management, and motivation. 

In sum, the findings of the K-means cluster analysis showed three main groups, i.e., three different main profiles of health care workers whom medical educators and risk managers meet in infection control in daily practice: (a)Amotivated and highly overestimated persons partially aware of their inadequate competencies and with a poor understanding of the risks of incorrect hand hygiene (“Unawares”);(b)Persons with moderate overestimation, intermediate performance, well-motivated, and appropriate risk assessment (“Recruitables”); and(c)Highly motivated persons with real confidence in their skills, above-average performance in self-assessed adherence to protocols, and a thorough understanding of the risks of incorrect hand hygiene (“Experts”).

**Unawares** working in critical processes may put patients at risk for in-hospital infections with a severe impact on patient outcomes [3] and the hospital economy [45,46,47] and may be highly resistant to educational interventions. This confirms our hypothesis that some persons overconfident in their competencies do not see any motivation to visit hygiene control programs: “*So, why should I visit tedious and time-consuming training on skills I already mastered and that for me have no visible effect on patients?*”. 

Extrinsic regulation by supervisors or by law (e.g., mandatory training) is needed to compel the unawares to visiting a training program to educate them. However, compulsory training (extrinsic motivation) is known to be suboptimal in competency acquisition [16] and thus may even be unethical from a deontological view. However, a utilitarian standpoint would demand compulsory training if effective learning methods and an appreciation for the potential risk of incorrect hand hygiene can thereby be created in those persons. However, it may be reasonable to assume that the learning dimension attitude [47] must be addressed and individually tested in these persons before further resource-consuming training in other domains (knowledge, psychomotor skills, or problem-solving) may be didactically, economically, and teleological conducted. 

**Experts**, on the other hand, are competent and confident but not overconfident. Therefore, it would be reasonable that they might not be motivated to attend IPC training. However, these participants not only showed competencies in hand hygiene, but they also had the attitude that repetitive learning is needed to prevent losing their competency. From the view of medical educators, this group comprises the goal of educational intervention and is easily addressed and motivated to participate in training. 

**Recruitables** may be an indistinct group with an assumed tendency to change to experts or unawares. They may be the main target group for medical educators because they are easily accessible. However, this stays an assumption, as our sample size was small and recruitables may be a more diverse group with variety amongst the different professional groups.

From an economical and didactic view, it seems reasonable that these three groups will profit from defining different learning objectives, which in turn demand diverse learning formats: Unawares have a low attitude and knowledge about the risks [16], demanding reflexive learning formats, while experts may lack problem-solving skills, e.g., speaking up [7,48]. Unfortunately, adaptive targeted learning for different and de facto large groups is time-consuming and requires a deeper understanding of the real proportions of these populations concerning hand disinfection and distractors such as workload [43]. 

According to our study, the cluster sizes may not indicate the accurate distribution of these clusters in different medical environments. This presents an opportunity for further examinations. It seems reasonable that those who are not motivated for hand hygiene and those who have a poor understanding of the risk of incorrect hand hygiene may not be motivated to participate in a study such as this. For more accurate evaluations, the entire staff of institutions or medical facilities needs to be studied to enhance generalizability and to test for distributions of the three clusters in different professions known for different compliance [7,49,50]. However, this study provides a first insight into the interaction of the different motivational effects.

### 4.3. Limitations 

Our study has several limitations. The non-probability sampling method used in this study limits the generalizability of the study results to the whole HCW population. It is possible that HCWs interested in infection control were also more motivated to participate in the study than those HCWs who were not interested in the topic. This is the common problem of open survey studies, known as self-selection recruitment and non-response bias [51]. Nevertheless, through the cluster analysis method, we also discovered the HCWs who were not motivated for infection control training, e.g., in the second identified cluster of the cluster analysis, and believe that this study can help understand attitudinal and behavioral patterns of overconfidence and motivation even in this subpopulation. 

Furthermore, the small sample size considerably limits our findings. However, we reached a sufficient sample size for performing the K-means algorithm regarding the number of the included variables [31]. Additionally, the K-means method is usually applied in large data sets, which opens the possibility for the cross-validation of the findings within the sample. Due to the small sample size, we could not perform cross-validation of the findings. Nevertheless, we did perform the external validation of the final cluster solution, comparing perceived patient safety levels across the clusters. The external validation results matched the previous findings in the literature [16,52]. The ceiling effects could have hindered the possible linear correlations between overconfidence and motivational variables [53]. The increase from five to, e.g., seven points of the Likert-scale of SIMS and SATIS should be considered in future research to reduce the skewness of the distributions. We included the self-reported hand disinfection compliance in the analyses, although it does not usually correspond to the actual hand disinfection performance, as reported in observational studies [9]. However, we included control items to restrict this perception bias as we asked the participants to assess how often they perform hand disinfection in situations that are not indicated by the WHO Guidelines. Additionally, we used the subjective self-estimation in hand hygiene compliance which can be flawed too as it is objectively usually lower than what is estimated by HCWs [54]. 

Moreover, the small sample size can be particularly sensitive to the influence of outliers. Thus, the results of quadratic regressions could have been affected by outliers. However, we cannot be certain that the outliers have been caused by measurement bias, such as entry errors. Furthermore, we believe that it is possible for outliers to correspond to actual participants, who were at the same time overconfident and amotivated. This extreme group of HCWs might exist in the praxis as we could demonstrate in the cluster analysis. Identifying and describing this group of HCWs is very important since this group might be resistant to interventions aiming to improve IPC competencies due to the lack of motivation to learn. On the other hand, this small group of extremely overconfident HCWs might be hypothetically responsible for outbreaks comparable to “superspreaders” in the pandemic. Furthermore, it is likely that a real proportion of this group could be even bigger due to the non-representative sampling method applied in the present study. Therefore, we decided not to remove these outliers from the analysis. Nevertheless, we applied the bootstrapping method to ensure robust estimation of standard errors and regression coefficients, which should not be significantly affected by outliers [55]. For precisely this reason, more work must be done on larger populations in future investigations. 

Another study limitation is that the data collection and analysis occurred before the outbreak of the COVID-19 pandemic. We believe that the hand disinfection practices could have changed in the meantime, which provides an opportunity for a post-pandemic retest. This is a cross-sectional study, wherein the interpretation of causal directions between the included variables is not feasible. Nevertheless, our assumptions were based on the previous findings, and we believe that our interpretation of the causal directions is theory-driven and, therefore, plausible.

## 5. Conclusions

We demonstrated in this study that the motivational process to attend ICP training is complex and associated with the HCWs’ own assessment of their own competencies, competencies of their colleagues, reported ICP practices, and knowledge about the risks for patients, when indicated hand disinfection is not provided. 

In particular, we were able to show that the motivation to learn in hand disinfection training is correlated in a u-shaped way to amotivation. However, the small proportion of the amotivational variance suggests that other variables play important roles in the motivational process in the context of IPC training and programs. One of these is the awareness of the risk to a patient after non-compliance with hand hygiene protocols. This is shown by the cluster analysis and one-way ANOVA results. In addition, it appears that the motivational process to actively participate in IPC training is complex and possibly influenced by a large spectrum of different variables. Considering how IPC training plays an important role in nosocomial infection prevention, future research should focus on identifying other motivational determinates. 

Further analysis of multi-professional data suggests that there are at least three groups of learners: legitimately confident and competent “experts” with preserved learning motivation, amotivated and incompetent “unawares” with high amotivation to attend training, and a broad group of “recruitables” with some overconfidence but also some learning motivation. Thus, overestimation and overplacement may have an impact on learning motivation. Furthermore, our hypothesis that overconfidence is correlated to poor learning motivation could be confirmed regarding unawares. However, formal interpretation of the direction of causal correlation is not permitted due to the study’s cross-sectional design.

These results might help medical educators to understand the diversity in mandatory training. Some persons are real experts with a high safety culture and a tendency for lifelong learning behavior; others were less motivated, and yet additional persons were amotivated and completely overconfident but incompetent according to used measurements. 

However, our results face limitations, such as small sample size and perhaps undetected subgroup effects, demanding further research. Nevertheless, our results may explain the diversity that medical educators, risk managers, and hygiene specialists face daily, sometimes causing frustrating results of extrinsically motivated mandatory training in infection control and possibly other disciplines (e.g., communication, data safety, and resuscitation).

This leads to the challenging situation that resource-consuming hygiene training with a known impact on patient safety, morbidity, and mortality may not work for some of the trainees due to low motivation, which places patients and the hospital economy at risk.

## Figures and Tables

**Figure 1 ijerph-19-05763-f001:**
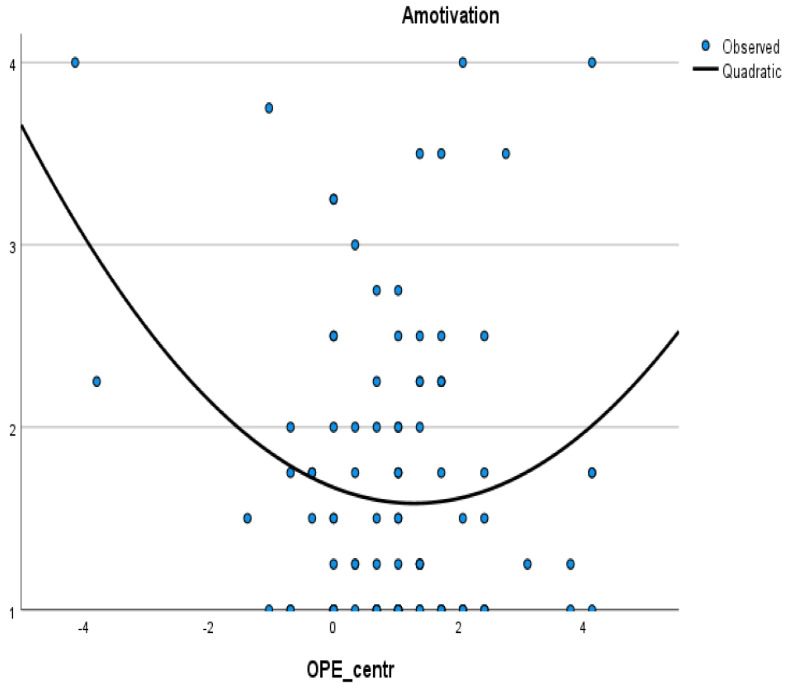
Graphical presentation of the quadratic association between overplacement (difference between self-assessment of own infection prevention and control (IPC) competencies and the competencies of the colleagues of the same profession) and amotivation. Note: OPE_centr-Overplacement is a centered variable (weighted through the scale mean value).

**Figure 2 ijerph-19-05763-f002:**
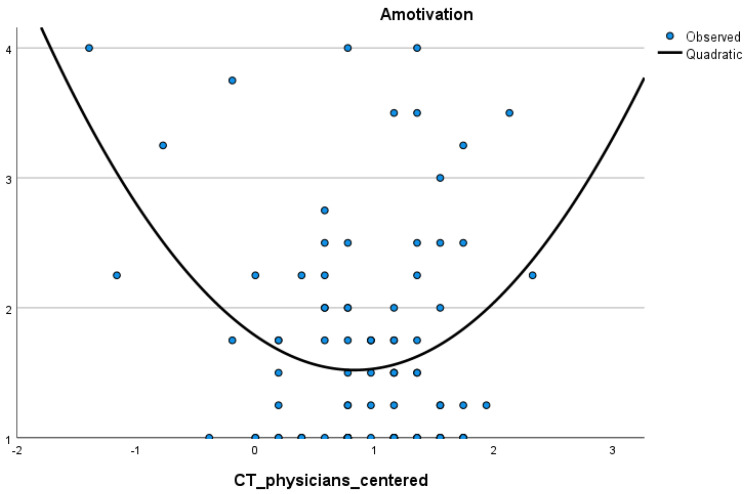
Graphical presentation of the quadratic association of clinical tribalism and amotivation. Note: CTE_physicians_centered represents clinical tribalism—a difference between nurses’ self-assessment and nurses’ perception of physicians’ infection prevention and control (IPC) competencies. CTE_physicians_centered is a centered variable (weighted through the scale mean value).

**Figure 3 ijerph-19-05763-f003:**
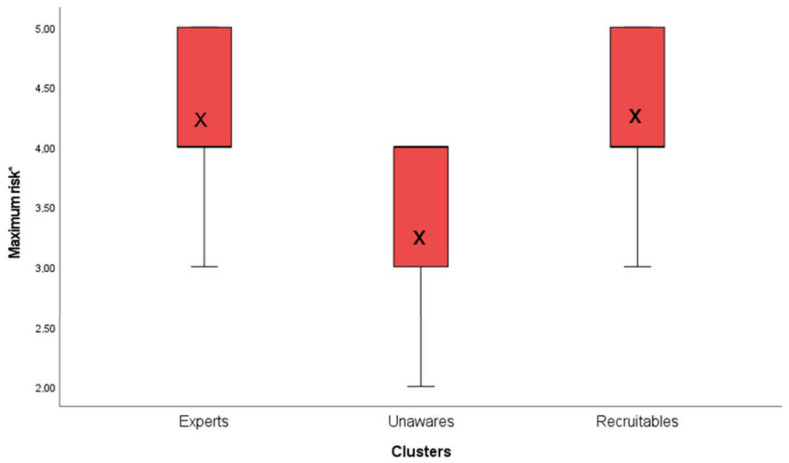
Boxplots of the variables associated with patients’ safety (the credible maximum risk of omitted hand disinfection) across all three clusters. Note: Maximum risk*—The item measures patients’ safety according to international standards of ISO 31000 of medical failure mode effect analysis (FMEA): “The maximum credible effect of omitted hand hygiene is: without consequence (1), minor—without any long-lasting effect (2), severe—with a prolonged hospital stay (3), critical—with long-lasting effects (4), and lethal (5)”; X—the mean value of the variable in the cluster. The one-way ANOVA determined the statistically significant difference between groups. Eta-squared showed 0.11 as the medium effect size. The Bonferroni post-hoc contrasts revealed that the maximum credible effect of omitted hand hygiene was estimated to be statistically significantly higher by the members of the first and third clusters in comparison to the second cluster.

**Table 1 ijerph-19-05763-t001:** Sample characteristics.

Characteristic		*n* (%)	Mean (Standard Deviation)
Age (years)		103	36.43 (17.31)
Gender	Male	21 (20.4%)	
	Female	79 (76.7%)	
	Divers/transgender/trans identical	1 (1%)	
Profession	Nurses	51 (49.5%)	
	Physicians	12 (11.7%)	
	Paramedics	3 (3.2%)	
	Others	36 (35%)	
Professional training	Completed	99 (93.2%)	
	Trainee	4 (3.9%)	
Workplace	Ambulant	7 (6.8%)	
	Primary hospitals		
	Secondary hospitals	17 (16.5%)	
	Tertiary hospitals	36 (35%)	
	(University hospitals)	19 (18.4%)	

**Table 2 ijerph-19-05763-t002:** Pearson’s product moment correlation matrix for hypothesis H1a.

Variable	Pearson Correlation *®*	Intrinsic Motivation	Identified Regulation	External Regulation	Amotivation	Self-Assessment	Overplacement	Clinical Tribalism Nurses vs. Paramedics	Clinical Tribalism Nurses vs. Physicians	Indicated Situations (WHO)	Non-Indicated Situations (WHO)
	*95% BCa* Confidence Interval										
Intrinsic motivation		*-*									
Identified regulation	*r* *BCa 95%*	0.77 ** {0.70–0.85]	-								
External regulation	*r* *BCa 95%*	−0.47 ** {−0.62–0.65]	−0.51 ** [−0.26–0.39]	-							
Amotivation	*r* *BCa 95%*	−0.55 ** [−0.68–−0.81}	−0.71 ** [−0.89–−0.59]	0.35 ** [−0.31–0.09]	-						
Self-assessment of own proficiencies	*r* *BCa 95%*	0.16 [−0.04–0.40]	0.12 [−0.34–0.28]	−0.09 [−0.31–0.09]	0.092 [−0.34–0.23]	-					
Overplacement	*r* *BCa 95%*	0.13 [−0.12–0.42]	0.12 [−0.14–0.48]	0.03 [−0.16–0.18]	−0.04 [−0.39–0.27]	0.48 ** [0.22–0.67]	-				
Clinical tribalism nurses vs. paramedics	*r* *BCa 95%*	0.03 [−0.19–0.25]	−0.04 [−0.22–0.26]	0.06 [−0.20–0.26]	0.03 [−0.17–0.23]	0.32 ** [0.06–0.46]	0.33 ** [0.17–0.50]	-			
Clinical tribalism nurses vs. physicians	*r* *BCa 95%*	0.10 [−0.12–0.36]	0.17 [−0.06–0.47]	0.12 [−0.07–0.26]	−0.11 [−0.44–0.16]	0.40 ** [0.06–0.50]	0.25 * [−0.06–0.50]	0.46 ** [0.27–0.63]	-		
Indicated situations (WHO)	*r* *BCa 95%*	−0.01 [−0.23–0.22]	−0.005 [−0.17–0.19]	−0.23 [−0.44–0.02]	0.16 [−0.05–0.34]	0.19 [−0.04–0.42]	−0.13 [−0.30–0.06]	0.11 [−0.09–0.29]	0.07 [−0.13–0.27]		
Non-Indicated situations (WHO)	*r* *BCa 95%*	−0.20 [−0.38–−0.02]	−0.10 [−0.30–0.09]	0.03 [−0.21–0.26]	0.16 [0.003–0.32]	0.24 [0.04–0.44]	−0.05 [−0.22–0.11]	−0.09 [−0.29–0.09]	0.08 [−0.10–0.24]	0.50 **	-

Note: ** Correlation is significant at the 0.01 level (two-tailed). * Correlation is significant at the 0.05 level (two-tailed). 95% (*BCa)* confidence interval and 95%- confidence intervals of Pearson’s coefficients after performing bias-corrected and accelerated (*BCa)* bootstrapping based on 1000 bootstrapped samples. Overplacement—the difference between the assessment of one’s own proficiencies and the assessment of the colleagues of the same profession. Clinical tribalism nurses vs. paramedics—the difference between the self-assessment of the proficiencies of the nurses and the assessment of paramedics’ proficiencies by nurses. Clinical tribalism nurses vs. physicians—the difference between the self-assessment of the proficiencies of the nurses and the assessment of physicians’ proficiencies by nurses. Indicated situations according to the World Health Organisation (WHO)—scenarios clearly defined as necessary for hand disinfection by WHO. Nonindicated situations (WHO)—scenarios where hand disinfection is not necessary according to WHO.

**Table 3 ijerph-19-05763-t003:** Results of the Bonferroni post-hoc comparisons after performing a one-way ANOVA for the dependent variable: “The credible maximum effect of omitted hand hygiene is without consequence (1), minor—without any long-lasting effect (2), severe—with longer hospital stay (3), critical—with long-lasting effect (4), and lethal (5)”.

Mean Values of the Dependent Variable in Clusters	(A) Cluster	(B) Cluster	(A)–(B) Difference between Means	*BCa* 95% Confidence Interval Lower	*BCa* 95% Confidence Interval Upper
Experts (*M*1 = 4.17)	Experts	Unawares	0.79 *	0.30	1.30
	Experts	Recruitables	−0.04	−0.41	0.34
Unawares (*M*2 = 3.38)	Unawares	Experts	−0.79 *	−1.27	−0.33
	Unawares	Recruitables	−0.83 *	−1.32	−0.41
Recruitables *M*3 = 4.20	Recruitables	Experts	0.04	−0.34	0.41
	Recruitables	Unawares	0.83 *	0.35	1.35

Note: Numbers in the colon “(A)-(B) Difference between means” correspond to differences of means between two groups. *BCa* 95 % Confidence Intervel lower and *BCa* 95 % Confidence Interval Upper are limits of the confidence intervals based on 1000 bootstrapped samples (bias-corrected and accelerated (*BCa*) bootstrap interval); * The difference is significant at the *p* < 0.05 value. ANOVA-Analysis of variance.

## Data Availability

The data presented in this study are available on request from the corresponding author.

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
