# Peer review of "Correlation between Overconfidence and Learning Motivation in Postgraduate Infection Prevention and Control Training"

_ijerph, 2022, doi:10.3390/ijerph19095763_

Round 1

Reviewer 1 Report

The manuscript entitled "Correlation between overconfidence and learning motivation in postgraduate infection prevention and control training" has a good idea in general but please consider the following comments:

1- The introduction is divided into subheadings, it is better to delete them.

2-  More precise data should be added to abstract

3- Please summarize the conclusion

Author Response

Dear reviewer. 

we thank for these suggestions and made the following changes to the manuscript basing on your feedback:  

1- The introduction is divided into subheadings, it is better to delete them.

We deleted the subheadings

2-  More precise data should be added to abstract

We added core results to the abstract

3- Please summarize the conclusion

We added a summarizing paragraph to the conclusion

Kind regards

SB

Reviewer 2 Report

Overall hypothesis is very well highlighted. Materials and methods are nicely explained and well connected. Questionnaire includes questions that are easily explained. I like the depiction of the results and further clasification of the subjects . The suthors have explained the categories of overconifdence based upon the questionnaire. 

I feel this study is a step in progress and I thank the authors for conducting a study in the middle of a pandemic. 

Author Response

Dear reviewer

we kindly thank you for your feedback!

Reviewer 3 Report

The authors have stressed out a critical problem that seems to be a minor one but which plays a essential role during patient treatment efficacy.

While the authors have done their best to carried this research, we notice few flaws that cannot be ignored:

  1. More than 1 self-citation to point out the same phenomenon. Please revise this and only cite the most relevant one.
  2. Would you mind to carry a statistical analysis between defined cluster in correlation with the various hospital levels (Primary, secondary, and tertiary)?

Author Response

Dear reviewer

thank your for your feedback on our manuscript. we made the following changes

More than 1 self-citation to point out the same phenomenon. Please revise this and only cite the most relevant one.

We included these different studies of our working group to highlight that the phenomonen occurs in different populations. However, we reduced the number of self-citations as suggested.

Would you mind to carry a statistical analysis between defined cluster in correlation with the various hospital levels (Primary, secondary, and tertiary)

We added a paragraph in the results section on this question. No differences between working places could be detected.

Kind regards 

SB

Reviewer 4 Report

This study demonstrated that OPE and CTE overconfidence effects were present and related to motivation to attend IPC training in this interprofessional sample of HCWs. The conclusions and limitations of this study were well described. The purpose of this study was clear and the implications were large, so the contribution to the body of knowledge was high. 

The abbreviation was written as the full name when it first appeared in the text, but there were parts that were difficult to read because there were too many similar abbreviations in the text. If possible, authors should refrain from using abbreviations.

Author Response

Dear reviewer

many thanks for your feedback on our manuscript. 

We checked all abbreviations and reduced their number for better understandability, especially concerning the different overconfidence effects.  

Kind regards

Stefan Bushuven 

Reviewer 5 Report

Trifunovic-Koenig et al., assessed the relationship between over placement/overestimation and motivation.  Although the concept, aims, and analyses of this study were quite simple and straightforward, the manuscript was written in a way it was difficult to read somehow. Here are some suggestions to make this manuscript read a bit easier.

  1. Introduction can be organized with more focus on logic, instead of focusing on defining the list of terms.
  2. Introduction: Add H1A and H1B in the subheading. Currently, there is no H1, and starts with H2 and H3.
  3. H2 and H3 should be combined in my opinion as H2 (clustering) is a necessary step prior to H3. They are not completely independent.
  4. Under Methods, the quantitative variable subheading is repetitive. I suggest revising the manuscript to organize each section better
  5. Results: why start describing supplementary data first? Start with the main finding presented in the main figure and tables then describe supplementary data to further support the main findings.
  6. Can authors provide the actual P-values instead of indicating them as p<.05? I am concerned about the quadratic regression result. Without those 2-3 outliers out of 103, I doubt it would make the significance.
  7. I see figures 1 and 2 then suddenly figure 5. Where are figures 3 and 4?
  8. As I commented above in 3, figure 5 can be moved to the supplementary. Instead, H3 results should be highlighted.

Author Response

Dear reviewer

thank you for your feedback on our manuscript. We made changes according to as follows: 

Although the concept, aims, and analyses of this study were quite simple and straightforward, the manuscript was written in a way it was difficult to read somehow.

We made some changes according to the feedback provided by the other reviewers. Additionally, we changed our manuscript as follows:

Introduction can be organized with more focus on logic, instead of focusing on defining the list of terms.

We had to decide between this suggestion and those of reviewer #2. In the past 3 articles on overconfidence, we had the “target audience” of infection control specialists and medical educators. In all three prior publications reviewers advised us to include a definition of the different effects as most readers might not be completely proficient with the different psychological effects. However, we made some changes to the introduction that might enhance readability.

Introduction: Add H1A and H1B in the subheading. Currently, there is no H1, and starts with H2 and H3.

We changed the manuscript according to this suggestion and moved tables from the supplemental part to the main manuscript.

H2 and H3 should be combined in my opinion as H2 (clustering) is a necessary step prior to H3. They are not completely independent.

We changed the manuscript according to this suggestion.

Under Methods, the quantitative variable subheading is repetitive. I suggest revising the manuscript to organize each section better

We changed the manuscript according to this suggestion.

Results: why start describing supplementary data first? Start with the main finding presented in the main figure and tables then describe supplementary data to further support the main findings.

We changed the manuscript according to this suggestion and re-arranged data presentation

Can authors provide the actual P-values instead of indicating them as p<.05? I am concerned about the quadratic regression result. Without those 2-3 outliers out of 103, I doubt it would make the significance.

We changed the manuscript according to this suggestion and made a statement to the phenomenon on the outliers. To our opinion it is not completely clear if these outliers are really outliners (bias) or extreme phenotypes. We included them as they might be an overlooked but maybe very important group. Hypothetically they may be comparable to superspreader effect in the pandemic. We added a paragraph focusing on this derived hypothesis.   

I see figures 1 and 2 then suddenly figure 5. Where are figures 3 and 4?

We corrected this.

As I commented above in 3, figure 5 can be moved to the supplementary. Instead, H3 results should be highlighted.

We discussed on this suggestion and those of reviewers #1,2,3 and 4 and decided to keep the boxplots in the main article for better visualization and understanding of the three clusters. However, we focused on the hypothesis H3 (now 2b) as suggested and gave more emphasis on the cluster analyses and the methods used.  

Round 2

Reviewer 5 Report

The manuscript has improved. However, there is no Figure 3 while there is a figure legend for Figure 3. Authors must make corrections.

Author Response

Dear reviewer 

thank you again for your work. 

we fixed some "typos" and checked figure 3.

This seems to be a technical problem (Word?). In our version the figure was present in the  document version with "markups" but not in the "without markups". We checked this technical issue again and hope that the recent version is correctly displaying the figure. 

kind regards